# OpenReview forum: "LLM Improvement for Jailbreak Defense: Analysis Through the Lens of Over-Refusal"
_NeurIPS.cc/2024/Workshop/SafeGenAi — SafeGenAi Poster_

### Official Review · Reviewer_1YqX · 2024-10-08
**[Accept] LLM Improvement for Jailbreak Defense: Analysis Through the Lens of Over-Refusal**

**Rating:** 8
**Confidence:** 3

**Review:**

## Summary:
The paper presents an innovative approach to addressing the challenge of defending large language models (LLMs) against jailbreak attacks while simultaneously minimizing the phenomenon of over-refusal, where LLMs inappropriately reject benign prompts. The authors propose and evaluate self-improvement and external-improvement strategies, providing a classification-inspired framework for assessing both harmful and harmless prompts. The study is well-motivated and contributes valuable insights into the balance between safety and usability in LLMs.

## Strengths:
* Relevance: The paper addresses a significant and timely issue in the deployment of LLMs - ensuring robust defense against jailbreak attacks without compromising their general utility. This is a highly relevant contribution to the field of AI safety and alignment.
* Innovation: The introduction of self-improvement and external-improvement as mechanisms to reduce jailbreak Attack Success Rates (ASR) without excessive model training is novel. The approach shows promise in reducing ASR to zero in several cases.
* Evaluation Framework: The authors propose a clear and methodologically sound evaluation framework that balances harmful and harmless prompts. This binary classification-style analysis is thorough and reveals key insights into over-refusal issues in existing defenses.
* Experimental Rigor: The use of the Llama-2 family of models for evaluation, along with state-of-the-art attack techniques, enhances the credibility of the findings. The experimental results are well-documented, with strong evidence for the effectiveness of the proposed strategies.

## Areas for Improvement:
* Over-Refusal Mitigation: While the paper thoroughly evaluates over-refusal, the proposed self-improvement and external-improvement strategies exhibit high over-refusal rates in certain contexts, particularly in in-context learning settings. The discussion acknowledges this but does not propose concrete solutions. Further exploration into reducing over-refusal without sacrificing jailbreak defense effectiveness would strengthen the paper.
* Lack of Generalization Across Models: The experiments are conducted solely on the Llama-2 family of models. It is unclear how well the proposed methods generalize to other LLMs. Broader applicability to different architectures (e.g., GPT-4, PaLM, etc.) should be tested or discussed.
* Keyword-Based Heuristic for ASR: The reliance on a keyword-based heuristic for measuring attack success rates may not capture the full complexity of model responses, especially in long-form, open-ended outputs. While the authors acknowledge this limitation, it would be helpful to explore more robust methods for determining whether a response is jailbroken.
* Impact on Instruction-Following: Although the paper reports that general instruction-following performance is maintained, the evaluation metrics used (InstructFollow) are acknowledged to have limitations, especially concerning false positives in scoring. A more robust metric or deeper qualitative analysis would provide stronger evidence of the method's impact on real-world tasks.
* Clarity in Methodology: The paper's description of the experimental setup could benefit from more clarity, particularly in explaining how prompts and improvements were structured. Detailed examples of the prompts and model responses would enhance understanding for readers.

## Overall Recommendation:
The paper is a valuable contribution to the field of LLM safety and jailbreak defense. It addresses an important and under explored problem which is over-refusal and offers novel solutions that show significant promise. However, improvements in addressing over-refusal rates and broader applicability would enhance the paper. Based on the strengths and the relatively minor weaknesses, I recommend **accept**

---

### Official Review · Reviewer_ECg1 · 2024-10-10
**This paper is relevant to the conference and I support the acceptance of this paper.**

**Rating:** 7
**Confidence:** 4

**Review:**

This paper proposes a method that reduces the over-refusal of language models while maintaining instruction following. The idea is novel and is based off the insight that we can ask an LLM to self-improve its response to avoid over-refusal. The paper illustrates how its approach substantially decreases the over-refusal rate while keeping instruction following, outperforming existing baselines. The paper also finds that existing defense methods have high over-refusal rates, validating the usefulness of the paper. Overall, this paper is highly relevant to the conference.

---

### Official Review · Reviewer_qy74 · 2024-10-10

**Rating:** 5
**Confidence:** 4

**Review:**

1. **Quality:**
The paper focuses on balancing robustness against jailbreak attacks and the potential over-refusal of benign prompts. The authors propose a model improvement mechanism that employs an LLM, whether internal or external, to transcribe outputs and enhance the security of LLM responses without extensive fine-tuning. However, this approach lacks technical contribution and clarity in several key areas:

   **a)** The proposed method merely uses the LLM to transcribe outputs, aiming to improve security, but it lacks a significant technical contribution. I am curious about how much improvement this transcription method offers compared to methods like Self Reminder [1], which directly integrates security cues into system prompts. Additionally, although few-shot improvement achieves the best defense results, it causes significantly more over-refusal than the baseline.

   **b)** The experimentation is insufficient. The main experiment only compares two types of token-level jailbreak attacks and lacks experiments on prompt-level jailbreak attacks [2], which are semantically based and therefore harder to defend against through transcription.

   **c)** The evaluation methods are biased. The keyword-based ASR evaluation tends to introduce significant bias. In contrast, evaluations performed by GPT-4o or humans might be more accurate. The authors calculate ASR and Over-Refusal rates based on keywords; could you also provide evaluations by GPT-4o?



[1] Xie, Yueqi, et al. "Defending chatgpt against jailbreak attack via self-reminders." Nature Machine Intelligence 5.12 (2023): 1486-1496.

[2] Chao, Patrick, et al. "Jailbreaking black box large language models in twenty queries." arXiv preprint arXiv:2310.08419 (2023).


2. **Clarity:**
The paper is generally well-organized, and the authors effectively use tables and figures to present their findings.


3. **Originality:**
Focusing on balancing ASR reduction and over-refusal is relatively novel. Most existing works concentrate solely on reducing ASR, overlooking how enhanced defenses might impede LLM utility by rejecting benign prompts. Nevertheless, the concept of using self-improvement strategies for LLMs isn't entirely new, and the proposed method lacks significant technical contributions.


4. **Significance:**
The contribution of this paper is valuable as it highlights the trade-offs between securing models and maintaining their usability. However, the actual impact might be limited by the lack of extensive validation across diverse LLM architectures and baselines. The authors could perform more experiments on different LLM architectures and could try using existing jailbreak attack benchmarks [1-2] to verify the effectiveness of their method.

[1] Chao, Patrick, et al. "Jailbreakbench: An open robustness benchmark for jailbreaking large language models." arXiv preprint arXiv:2404.01318 (2024).

[2]Xu, Zhao, Fan Liu, and Hao Liu. "Bag of Tricks: Benchmarking of Jailbreak Attacks on LLMs." arXiv preprint arXiv:2406.09324 (2024).